# The Fuel of Our Future: Hydrogen or Methane?

Vladimir Arutyunov [1,2,*] , Valery Savchenko [2] , Igor Sedov [2] , Artem Arutyunov [1] and Aleksey Nikitin [1,2]

1    N.N. Semenov Federal Research Center for Chemical Physics, Russian Academy of Sciences, Kosygina 4, 119991 Moscow, Russia; aarutyunovv@gmail.com (A.A.); ni_kit_in@rambler.ru (A.N.)

2    Institute of Problems of Chemical Physics, Russian Academy of Sciences, Semenova 1, 142432 Chernogolovka, Russia; savch1152@mail.ru (V.S.); isedov@icp.ac.ru (I.S.)

*    Correspondence: v_arutyunov@mail.ru

**Abstract:** Growing concern about climate processes has caused an interest in low-carbon fuels, such as methane and hydrogen. Although hydrogen seems to be beyond comparison in this regard, the need for high energy consumption for its production—mainly due to the same fossil hydrocarbons, low specific volume energy, and problems with its storage and transportation—make the production and consumption in the "hydrogen energy" chain extremely expensive, and even environmentally unattractive. Estimates show that it is significantly inferior to methane-based energy not only in terms of costs and efficiency, but also in terms of global $CO_2$ emissions. The vast resources of natural methane, primarily gas hydrates, are able to provide humanity with energy and hydrocarbons for hundreds of years. Meanwhile, promising modern technologies for the conversion of methane into basic chemicals—including new autothermal technologies for its oxidative conversion into syngas and its direct conversion into chemicals—allow the consideration of methane not only as a fuel, but also as the basis of future organic chemistry. Methane and other hydrocarbons, synthesized using thermonuclear energy from $CO_2$ and water—which are abundant on the Earth—can remain the most convenient mobile, easily stored and transported fuels and universal chemical raw materials, even after the inevitable transition to thermonuclear energy in the distant future. The inclusion of $CO_2$ through the synthesis of methane into the global energy cycle will allow real global carbon neutrality to be achieved.

**Keywords:** climate change; global energy; natural gas; methane; gas chemistry; conversion of methane; syngas; matrix conversion

## 1. Introduction

Growing concern about climate change and the already-obvious impossibility of solving the problem through renewable energy sources alone [1,2] has caused a surge in interest in low-carbon fuels. Of course, with regard to $CO_2$ emissions, hydrogen is beyond comparison. Therefore, the Paris Climate Agreement [3] in 2015 proclaimed the transition to hydrogen as the main direction of combating climate change.

Although the real need and even the fundamental possibility of managing global climate processes raise many questions [4], the trend of transition to low-carbon energy has contributed to the popularization of the concept of "hydrogen energy", and intensified research on the use of hydrogen as a fuel [5]. According to forecasts, annual investments in hydrogen technologies over the period 2018–2022 are expected to reach EUR 1.9 billion per year, and by 2050 global hydrogen consumption may reach 370 million tons per year [6].

In the long term, after the inevitable development of thermonuclear fusion energy by mankind, hydrogen—the oxidation of which produces only water—has every chance of becoming a universal environmentally friendly energy carrier. However, the current transition to "hydrogen energy" is associated with very serious economic, energy, and infrastructure problems [7,8], so its practical implementation before the development of thermonuclear energy is very doubtful. At present, the conversion of natural gas is

practically the only source of hydrogen that is acceptable both economically and from the point of view of the availability of necessary resources. All renewable sources of energy or those based on carbon capture and storage (CCS) technologies have insufficient global capacity, and are extremely expensive. They are also environmentally unattractive if all carbon emissions during the full life cycle of the production of appropriate equipment and subsequent hydrogen production are objectively taken into account [7,8]. Estimates show that hydrogen is several times inferior to natural-gas-based energy not only in terms of costs and efficiency, but also in terms of global $CO_2$ emissions during the actual hydrogen production chain. This is why efforts are currently being made in several European countries to legally classify natural gas and nuclear power as "green" energy sources. In reality, the vast resources of various natural sources of methane—primarily gas hydrates—are sufficient to provide humanity with the necessary energy and hydrocarbon raw materials for many dozens and even hundreds of years. Thus, until industrial-scale thermonuclear fusion is mastered, it will be impossible to provide humanity with the necessary amount of energy without them [1].

At present, the most effective way to reduce the carbon footprint of the global economy is to increase the efficiency of technologies for converting natural gas (methane) into chemical products and motor fuel. However, even in the distant future, methane and other hydrocarbons synthesized using energy from thermonuclear fusion from the excess of $CO_2$ and water available on the Earth can remain the most convenient, mobile, easily stored and transported energy carriers and universal raw materials for organic chemistry. The inclusion of $CO_2$ through the electrochemical synthesis of methane and other hydrocarbons in the global energy cycle will allow real global carbon neutrality to be achieved.

Currently, most technologies for the chemical processing of methane are based on traditional catalytic processes, which are highly complex and energy-intensive. The aim of this paper is to consider some modern and promising autothermal non-catalytic technologies for converting methane into basic chemicals, including those for the oxidative conversion of methane into syngas, and for its direct conversion into other chemicals.

## 2. Conversion of Methane to Syngas: Current Significance and Possible Role in the Future

Of course, the leading gas chemical technology today is the conversion of natural gas into syngas. The uniqueness of syngas for gas chemistry is that it is practically the only product into which methane can be converted completely and in a thermodynamically equilibrium process. This process is the first stage in the production of almost all large-tonnage gas chemical products, such as ammonia, methanol, hydrogen, and synthetic crude oil [9–11]. However, it should be noted that the conversion of thermodynamically highly stable methane requires very high temperatures of up to 1000 °C and, therefore, a lot of energy. Currently, the cheapest and most affordable energy source for this is the oxidation of methane itself, which makes autothermal gas chemical processes particularly attractive. No renewable sources can provide the necessary amount of sufficiently inexpensive energy for the industrial conversion of methane into syngas. Only new-generation high-temperature gas-cooled nuclear reactors (HTGRs) [12] are currently considered as an alternative energy source for methane conversion. However, the potential of nuclear power is limited by insufficient reserves of uranium in the Earth's crust [13], preventing it from seriously expanding its share in the global consumption of energy.

Thus, despite all the complexity and energy intensity of modern natural-gas-to-syngas conversion processes, among which catalytic steam–methane reforming (SMR) occupies a leading place, they remain the basis of modern gas chemistry and the main, if not the only, real industrial source of hydrogen. This complex, energy-intensive, and capital-intensive process consumes up to 70% of the cost of the target gas chemical products. Specific capital expenditures at the largest "world class" gas chemical enterprises, with an annual capacity of over 1 million tons, reach USD 200,000 per barrel per day. This is significantly higher than for oil refineries or petrochemical enterprises, and seriously hinders the development of gas chemistry. Therefore, numerous attempts are currently being made to develop simpler

and more energy-efficient processes for converting methane into syngas, which would contribute to a wider use of vast unconventional methane resources. This could also be the most realistic way to reduce global $CO_2$ emissions when using and processing natural gas.

Currently, many companies and research groups around the world are actively searching for more simple and efficient methods of converting methane into syngas. Among them are such interesting processes as oxidation on ceramic membranes, processes based on short-time catalysis, the use of microchannel and membrane reactors, production of syngas employing power plants such as diesel engines and gas turbines, oxidation of methane with metal oxides, filtration combustion, and others [14–17]. However, despite intensive efforts and a number of sufficiently large pilot installations, none of these methods has yet reached the industrial level.

Autothermal methods of methane-to-syngas conversion are attractive not only because they use such a cheap energy source as methane itself, but also due to the absence of heat losses during heat transmission and expensive metal-intensive heat-exchange equipment, ensuring the lowest energy consumption and capital costs. Non-catalytic processes have additional advantages, since not only there are no capital and operating costs for the catalyst and its regeneration, but also the requirements for the composition and preliminary preparation of the raw gas are significantly reduced. Below, we consider one of the most promising new methods of conversion of hydrocarbon gases into syngas from our point of view—so-called matrix reforming (MR).

It is important that, in the foreseeable future, with the implementation of thermonuclear fusion energy, the cheapness and excess of unconventional methane can allow much wider use of the conversion of its huge resources to produce syngas and, thus, synthetic methanol or Fischer–Tropsch products (e.g., synthetic liquid hydrocarbons, syncrude oil). It should also be noted that after the development of thermonuclear fusion energy, the global role of methane and the processes of its conversion into chemicals is likely to remain even after the exhaustion of industrially significant hydrocarbon resources on our planet. The inclusion of $CO_2$ through the electrochemical or catalytic (and, in the longer term, possibly biological) synthesis of methane [18,19] or methanol [20] in the global energy cycle (Figure 1) will make it possible to truly achieve global carbon neutrality.

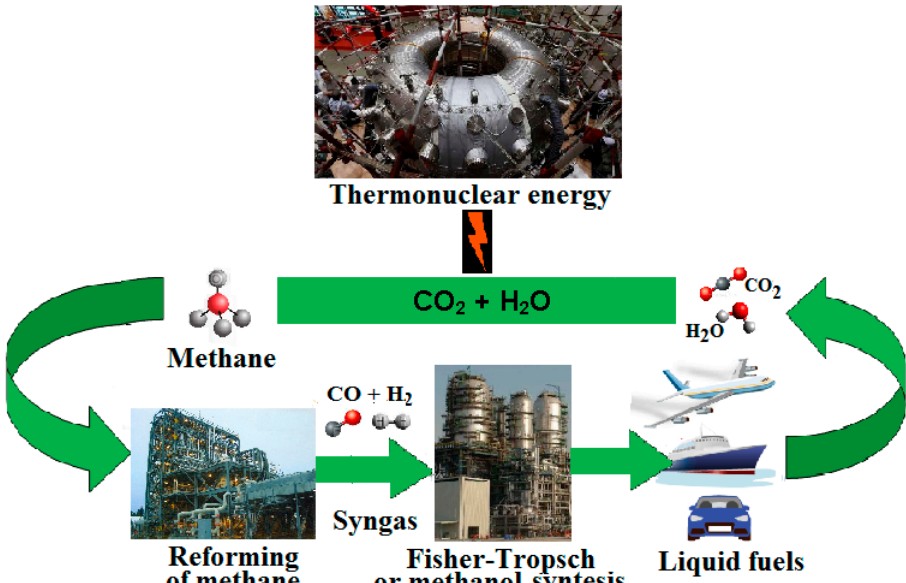

**Figure 1.** An energy cycle based on thermonuclear energy and methane, which allows real global carbon neutrality.

The use of electrochemical $CO_2$ reduction to produce fuels or value-added chemicals by using excess energy has attracted significant attention as a means of reducing the amount of $CO_2$ in the atmosphere and storing electrical energy. It has been demonstrated that

methane can be selectively generated on carbon-supported Pt nanoparticle catalysts with notable efficiency [18]. The ruthenium core–shell metal/carbide nanoparticles are able to catalyze $CO_2$ hydrogenation at low temperatures (160−200 °C), with selectivity to methane of up to 100%, far surpassing the most active Ru catalysts [19], which makes them very promising candidates for producing methane in an abundance of carbon dioxide and water via methanation of $CO_2$ using the Sabatier reaction:

$$CO_2 + 4H_2 \rightarrow CH_4 + 2H_2O \rightarrow \Delta H^0{}_{298} = (-253.2 \text{ kJ/mol}) \tag{1}$$

Thus-obtained methane can be used as a gas fuel or converted into syngas for further production of liquid hydrocarbons. Another possibility is the direct $CO_2$-based synthesis of methanol [20] (Figure 2):

$$CO_2 + 3H_2 \rightarrow CH_3OH + H_2O \rightarrow \Delta H^0{}_{298} = (-49.2 \text{ kJ/mol}) \tag{2}$$

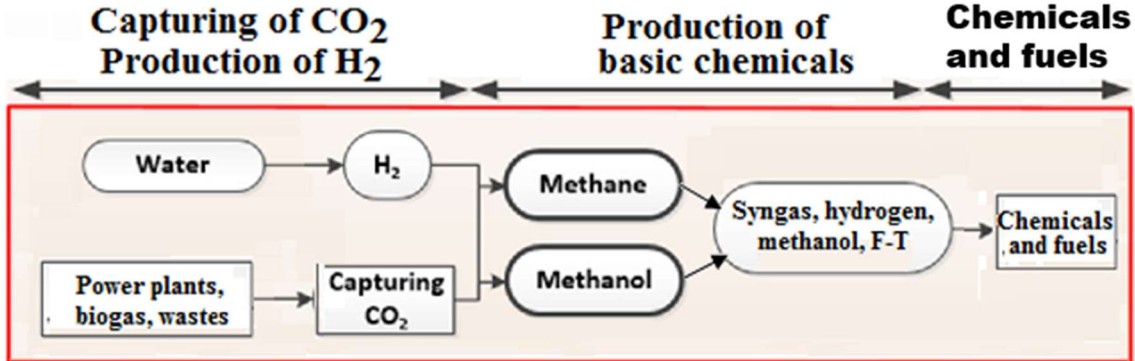

**Figure 2.** Possibility of methane production via the Sabatier reaction of $CO_2$-based methanation and direct $CO_2$-based synthesis of methanol.

Although currently $CO_2$-based methane and methanol do not compete with traditional technologies of their production (the estimated cost of production of methane based on $CO_2$ is 332−493 ct/kg, while that of methanol is 136−211 ct/kg [20]), this situation may change in the future, and the resulting methane, if necessary, can be used to produce hydrogen via steam–methane reforming:

$$CH_4 + 2H_2O \rightarrow 4H_2 + CO_2 \rightarrow \Delta H^0{}_{298} = (-226 \text{ kJ/mol}) \tag{3}$$

where $CO_2$ is recuperated as a concentrated product that can be easily converted back to methane.

Thus, the significance of methane for global energy lies not only in the fact that it is the cheapest and most environmentally friendly energy carrier of our time, but also in the fact that it has every chance to maintain this position in the distant future.

### 3. Matrix Reforming of Natural Gas to Syngas

As already noted above, the basic principles that, in our opinion, should be inherent to simple and efficient processes of reforming natural gas into syngas are the autothermal and non-catalytic nature of the processes. Such conditions are met by superadiabatic processes, such as filtration combustion [14,17] and matrix reforming [21–23], wherein part of the heat of the hot conversion products is transferred to fresh reagents.

The basic principles of the matrix reforming are quite simple, and are described in more detail in [21]. The process is based on a weakly exothermic partial oxidation of methane:

$$CH_4 + 0.5O_2 \textcircled{R} CO + 2H_2 \rightarrow \Delta H^0{}_{298} = -36 \text{ kJ/mol} \tag{4}$$

However, the heat release in this process, which requires an oxygen excess coefficient $\alpha = [O_2]/2[CH_4] = 0.25$, is too small for its stable implementation. Therefore, it is inevitably accompanied by a parallel reaction of complete oxidation of methane:

$$CH_4 + 2O_2 \,^{\circledR}\, CO_2 + 2H_2O \rightarrow \Delta H^0{}_{298} = -802 \text{ kJ/mol} \tag{5}$$

which provides the necessary additional amount of heat. The goal of any technology based on the partial oxidation of methane to syngas is to minimize the contribution of the complete oxidation reaction in order to increase the selectivity of syngas formation and approach the value $\alpha = 0.25$.

Sustainable conversion of very rich mixtures of hydrocarbons with an oxidizer is ensured by the recovery of part of the heat of the products into a fresh mixture of reagents. During filtration combustion, this is usually provided by heat transfer through a solid heat carrier, with the countercurrent movement of the gas flow relative to it. Since filtration combustion, as well as all combustion processes in porous media, proceeds with a complex interaction of solid and gas phases, it is more suitable for conversion of solid fuels and waste to syngas.

In matrix conversion, the process proceeds entirely in the gas phase, near the surface of a gas-permeable solid matrix. Heat transfer from conversion products to fresh reagents occurs through the convectively and radiatively heated gas-permeable walls (matrix) that form the reactor cavity. Intensive convective and radiation heat recovery from conversion products to fresh reagents, along with the locking of the IR radiation of the flame front during flameless combustion near the inner surface of the permeable volumetric (3D) matrix, significantly expand the combustion limits, and ensure the possibility of operating with values of oxygen excess coefficient as low as $\alpha = 0.34$–$0.36$ [21]. Figure 3 shows the inner construction of the matrix reformer and its overall view. The flows of the hydrocarbon gas and the oxidizer (atmospheric air, enriched air, or oxygen), in some cases with the addition of steam, enter the mixer, and then symmetrically from both sides into the cylindrical matrix reformer. After conversion near the inner surface of two symmetrically arranged flat round matrices (6) fabricated from pre-pressed FeCrAl alloy (Fehral) wire, the resulting syngas exits through the upper outlet pipe (3). The radiation screen (7) provides an additional reflection of the radiation energy of the flame front on the matrix.

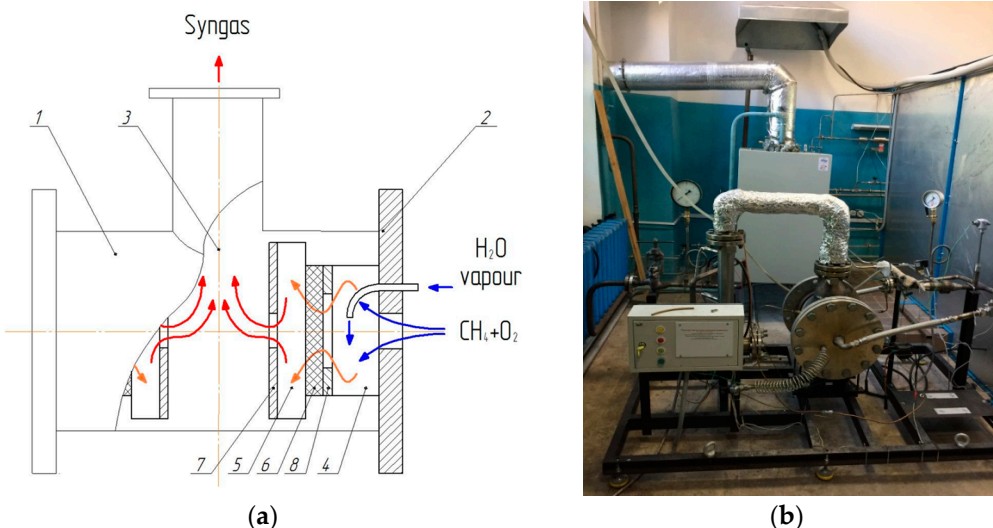

(**a**) (**b**)

**Figure 3.** (**a**) Schematic diagram of the matrix reformer and (**b**) the appearance of a demonstration matrix reformer with a capacity of up to 10 m$^3$/h for incoming gas for operation at a pressure of up to 10 atm: (1)—core vessel; (2)—flange; (3)—outlet pipe; (4)—mixing chamber; (5)—combustion chamber; (6)—matrix; (7)—radiation screen; (8)—inlet aperture.

Demonstration reformers with a capacity of up to 10 m$^3$ of incoming gas show the advantages of the process. Operation with atmospheric air allowed us to obtain a stable conversion at $\alpha$ = 0.34–0.36, providing a concentration of H$_2$ up to 25% and of CO up to 14% in the resulting syngas. The concentration of CO$_2$ was ~3.6%, and the rest was nitrogen. As the oxidizer was enriched with oxygen, the concentrations of the syngas components increased proportionally, and the nitrogen concentration decreased. The use of air enriched to an appropriate degree, with oxygen as an oxidizer, can allow one-stage production of syngas with the composition necessary for the synthesis of ammonia.

Operation with technical oxygen allowed us to obtain a concentration of H$_2$ up to 54%, and a concentration of CO up to 31%, with the H$_2$/CO ratio in the range of 1.6–1.8. The conversion of both reagents exceeded 95%.

The main advantages of the matrix conversion, which make it very promising—especially for the rapidly developing low-tonnage natural gas conversion (for example, for low-tonnage production of hydrogen directly in the places of its consumption), are as follows [22]:

- It is autothermal process needs no additional heat or energy;
- The process allows processing of hydrocarbon gases of almost any composition, including those with a high content of methane homologues, associated petroleum gases, refinery gases, biogas, etc.;
- The process provides the possibility of a wide range of possible capacities, including small-tonnage installations;
- The process is very compact. Its specific volume capacity is at least 10 times higher than that of steam reforming, which significantly reduces capital costs;
- The process is quite simple in design and operation, which significantly reduces operating costs;
- The absence of a catalyst significantly decreases the demand for the purification of the feedstock.

The possibility of safe and simple production of inexpensive syngas based on matrix conversion of hydrocarbon gases by atmospheric air opens real prospects for the firect small-tonnage conversion of natural and even associated petroleum gases under field conditions [23]. As was shown by semi-industrial tests of the Fischer–Tropsch synthesis using inexpensive syngas obtained via the oxidation of methane with atmospheric air [24], such a process not only significantly reduces the cost of the synthetic hydrocarbons obtained, but also has a number of additional advantages [25]. Especially important are the possibility of a significant reduction in capital costs and the possibility of safe operation in the field conditions, since high CAPEX and the use of oxygen in the field conditions are practically excluded. It is worth noting that the composition of syngas used in [24,25] and a number of other studies broadly coincide with the composition of syngas obtained via the matrix conversion of natural gas by atmospheric air.

Matrix conversion allows the utilization of low-resource sources of inexpensive and easily obtained biogas and other renewable hydrocarbon gases for the production of liquid biofuel through their air conversion into low-cost nitrogen-rich syngas, followed by Fischer–Tropsch synthesis in a cascade of sequential reactors [26].

Low-tonnage products obtained in this way may include hydrogen. The low-tonnage distributed production of hydrogen directly in the places of its consumption allows by-passing of the problems of storage and transportation of its large volumes, which do not yet have practically acceptable solutions. To increase the yield of hydrogen, the gas-phase process of matrix reforming can be supplemented by a sequential catalytic process of steam conversion of the formed carbon monoxide—a so-called water–gas shift reaction (WGSR). The autothermal nature of matrix reforming allows a substantial reduction in the carbon footprint compared to steam reforming.

Matrix reforming opens up the possibility of small-tonnage production of universal and popular products such as methanol, which is a universal fuel and one of the most efficient liquid hydrogen carriers. Our experiments with syngas of the composition ob-

tained through matrix reforming using atmospheric air have shown the possibility of stable catalytic synthesis of methanol with a CO conversion per pass through the reactor of at least 20%, and a selectivity of methanol formation of up to 95%. At the same time, fuel gas consumption is reduced by a factor of ~4 in comparison with the traditional industrial processes, water consumption is reduced by ~5-fold, and capital costs are significantly reduced.

The great importance of increasing the syngas yield during matrix reforming is the optimization of the processes in the post-flame zone after the complete conversion of oxygen. The kinetic simulation of the process revealed three characteristic stages of hydrocarbon matrix conversion (Figure 4). In flame zone I, in the lack of oxygen, along with CO, $H_2$, $CO_2$, and $H_2O$, the products of hydrocarbon pyrolysis are formed. Then, in the post-flame zone II, in the absence of oxygen, at 1400–1600 K, a slow pyrolysis of residual hydrocarbons occurs together with steam and $CO_2$ conversion of products. Kinetic analysis shows that the pyrolysis of hydrocarbons into acetylene and the subsequent steam and dry reforming of the latter proceeds much faster than the direct interaction of hydrocarbons with $H_2O$ and $CO_2$, leading to a decrease in the concentrations of hydrocarbons and acetylene and an increase in the $H_2$ and CO concentrations. The final composition of the products is established in zone III, in which they slowly approach thermodynamic equilibrium. However, at the reforming temperature, the time required for this significantly exceeds the time that they stay in the reformer. Therefore, the optimization of conditions in the post-flame zones II and III allows decreasing the acetylene yield, while increasing the syngas yield [27,28].

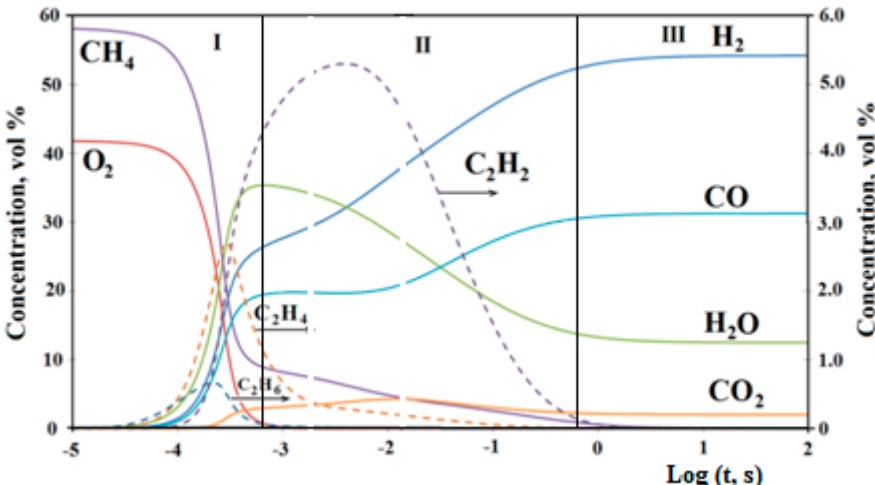

**Figure 4.** Calculated distribution of reagents and products in the matrix reformer at $O_2$: $CH_4$ = 0.72: 1, $T$ = 1700 K.

The processes of steam and dry reforming of hydrocarbons in the post-flame zones make it reasonable to introduce an additional amount of steam and/or $CO_2$ into these zones to increase the syngas yield. In the case of the introduction of an additional amount of carbon dioxide—for example, by its recycling after separation from the resulting syngas—such a process can be considered as a method of partial utilization of $CO_2$. According to kinetic modeling, the non-catalytic dry reforming of hydrocarbons in the post-flame zones begins with their thermal pyrolysis, followed by the conversion of $CO_2$ via interaction with radicals formed during pyrolysis. At the same time, the conversion rate of the residual hydrocarbons does not depend on the concentration of the supplied $CO_2$ [29].

## 4. Direct Technologies for the Conversion of Methane into Chemicals

Direct conversion of methane into chemicals has been the Holy Grail of researchers for more than a century. The main problem that has to be overcome in this direction is the high thermodynamic stability of methane, which exceeds the stability of almost all of the target products of its conversion. Therefore, it is difficult to expect a solution to

this problem within the framework of equilibrated catalytic processes. A striking and most popular example of an attempt to create a direct methane conversion process is the catalytic process of oxidative coupling of methane (OCM) into ethane and ethylene [30,31]. Despite many years of effort and a huge number of publications on this process, it has not yet been commercialized. The main problem is the need for a high rate of conversion of methane in one form or another into methyl radicals, at a high concentration of which their gas-phase recombination into ethane occurs, followed by rapid conversion of the latter into ethylene—the basic product of modern petrochemistry—which under these conditions in the gas phase is more stable. Unfortunately, the rate of oxidation of ethylene itself, both in gas-phase processes and on the surface of the catalyst used to generate methyl radicals, is too high, which limits the yield of ethylene to less than 25% [32]. This is not enough for the process to be commercially attractive [31]. Modern physical methods of generating methyl radicals, such as plasma chemical methods, require too much energy, so at present they are unprofitable. However, it can be assumed that after the implementation of thermonuclear fusion energy, an abundance of inexpensive energy will make the direct plasma chemical processes—both oxygen-free and oxidative condensation of methane into ethane, ethylene, and heavier hydrocarbons—more economically attractive.

In the second half of the last century, the idea of methane functionalization based on its preliminary halogenations was very popular. A number of processes are being developed at the pilot plant level [33,34]. Recently, the process of partial oxidation of methane into acetylene, which has long been an industrial process, has again attracted attention [35–37]. Computer modeling of the process shows the possibility of achieving almost 40% selectivity of acetylene formation at the conversion of hydrocarbons above 80% [35]. The molar fraction of $C_2H_2$ at the outlet can reach approximately 10% under industrial-scale production conditions [35]. It is possible to foresee a bright future for this process, especially in the presence of affordable and inexpensive energy, since it allows obtaining a sufficiently high yield of such demanded and chemically versatile products as acetylene directly, in one stage, via a fairly well-developed technological process.

One of the oldest technologies for direct conversion of methane is direct methane oxidation to methanol (DMTM) [38]. This process was of industrial importance in the early period of the development of petrochemistry, but then gave way to more efficient processes based on syngas. The main problem of the process is the relatively low conversion of methane per pass, even in a sectional reactor with a separate supply of oxidizer to each section, amounting to only 6–8%, with the selectivity of methanol formation at a pressure of 70–80 atm being ~50% [39]. The low selectivity for methanol is due to the branched-chain gas-phase mechanism of this process, with very rapid branching and a short chain length. This causes the failure of numerous attempts to increase its effectiveness through the use of catalysts or various physical methods of promotion [38].

The yield of methanol can be increased by recycling the outgoing gas, but this requires the use of expensive oxygen or oxygen-enriched air, which complicates and increases the cost of the process. Nevertheless, the simplicity of the process and the ability to convert almost any hydrocarbon gases into methanol, including associated petroleum gas, makes this process still attractive, especially in combination with the ability to use outgoing gas to generate electricity [38,39]. It should be noted that CO formed during this process is also an important basic product of gas chemistry. Therefore, the carbon selectivity of the process can be significantly increased if this second most important carbon component of the DMTM is used to carbonylate the resulting methanol into acetic acid [40].

Currently, in connection with the search for ways to reduce the carbon footprint of energy through the use of hydrogen, methane pyrolysis into hydrogen and solid carbon,

$$CH_{4(g)} \,^{®}\, C_{(s)} + 2H_{2(g)} \rightarrow \Delta H^0{}_{298} = 74.85 \text{ kJ/mol of } CH_4 \qquad (6)$$

both catalytic and non-catalytic, has received special attention to produce $CO_2$-free $H_2$ [41–43]. Due to the impossibility, already discussed above, of ensuring the volume of hydrogen production necessary for the global energy sector [1,7]—including methane pyrolysis, from

renewable sources only, as well as the inevitable problem of annual sequestration of up to 5 billion tons of fine carbon generated via this process—it is difficult to expect widespread use of this technology. However, in the future, with the advent of thermonuclear energy, it is possible to expect an increase in its significance.

## 5. Conclusions

Currently, methane is the most affordable, mobile, and environmentally friendly energy resource on our planet, with which neither renewable energy sources nor hydrogen can compete on the global scale. The global pipeline network and the liquid methane transportation system have already made it a global energy carrier capable of surpassing oil in value in the near future. The existing processes of conversion of methane into syngas and other products already make it possible to obtain from it almost everything that is produced from oil, and there are good prospects for further improvement and increasing the efficiency of these processes.

However, it should be noted that the importance of methane is not limited to the role of convenient fuel for the immediate future of the global economy. It is important that, in the foreseeable future, with the development of thermonuclear fusion energy, the cheapness and excess of unconventional methane can allow much wider use of the conversion of its huge resources to produce syngas and, thus, methanol and synthetic Fischer–Tropsch products (e.g., synthetic liquid hydrocarbons, syncrude oil)— very convenient liquid fuels. It should also be noted that after the development of thermonuclear fusion energy, the global role of methane and the processes of its conversion into chemicals are likely to remain even after the exhaustion of industrially significant hydrocarbon resources on our planet. The inclusion of $CO_2$ through the electrochemical synthesis of methane and other hydrocarbons in the global energy cycle will make it possible to truly achieve global carbon neutrality. Thus, the significance of methane for global energy lies not only in the fact that it is the cheapest, most environmentally friendly energy carrier, but also in the fact that it has every chance to maintain this position even in the distant future. The inclusion of $CO_2$ through the synthesis of methane into the global energy cycle will allow real global carbon neutrality to be achieved.

**Author Contributions:** Conceptualization, V.A.; methodology, V.A., V.S. and I.S.; investigation, V.A., V.S., A.A. and A.N.; writing—original draft preparation, V.A., V.S. and I.S.; writing—review and editing, V.A., V.S. and I.S.; project administration, I.S. All authors have read and agreed to the published version of the manuscript.

**Funding:** This research was performed within the framework of the Programs of Fundamental Research of the Russian Academy of Sciences on the research issues of FRCCP RAS No. 0082-2019-0014 (State reg. AAAA-A20-120020590084-9) and IPCP RAS No. 0089-2019-0018 (State reg. AAAA-A19-119022690098-3).

**Institutional Review Board Statement:** Not applicable.

**Informed Consent Statement:** Not applicable.

**Data Availability Statement:** Not applicable.

**Acknowledgments:** The experimental part of this work was performed using the equipment of the Center for Collective Use "New Petrochemical Processes, Polymer Composites, and Adhesives" IPCP RAS (no. 77601).

**Conflicts of Interest:** The authors declare no conflict of interest. The funders had no role in the design of the study; in the collection, analyses, or interpretation of data; in the writing of the manuscript, or in the decision to publish the results.

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
