# Peer review of "The Fuel of Our Future: Hydrogen or Methane?"

_methane, doi:10.3390/methane1020009_

Round 1

Reviewer 1 Report

1.      In line 46 (page 2), the authors have highlighted that natural gas is practically the only source of hydrogen that can make a noticeable impact on CO2 emissions by the global energy sector.  However, many other alternatives are available for hydrogen sources, such as ammonia, biomass gasification, etc. Hydrogen can be produced by ammonia cracking or pyrolysis, and it will also not produce CO2 (Due to the non-avalibity of carbon in ammonia). The authors need to give more explanation about this.

2.      Authors seem to mainly focus on natural gas as a hydrogen alternative. However, other good alternatives are also available for hydrogen production. Other alternatives also need to highlight in the manuscript.

3.      The author needs to do an extensive study about the advantages and disadvantages of traditional and developing technologies for hydrogen production from different sources. Can add a table of comparison because it will give a different benchmark for comparison.

4.      The Sabatier reaction or Sabatier process produces methane and water from a reaction of hydrogen with carbon dioxide (CO2 + 4H2 = CH4 +2H2O). However, authors have written the reaction as CO2 +4H2 = H4 +2H2O, needs to correct the chemical reaction.

5.      The author should include a table or graph of the cost and CO2 footprint of hydrogen production from the different processes such as coal gasification, ammonia pyrolysis, steam reforming of methane, water electrolysis, biomass gasification, and other available processes.

6.      The sentence formulation in the manuscript is not done correctly and has many grammatical mistakes in lines 66, 84,142,157,163,270,278,315,326.

Author Response

We are grateful to the Reviewer for the careful analysis of our work and tried to take into account the comments made as much as possibl. Please see the attachment. 

Reviewer 2 Report

The paper represents an interesting point of view on the possible development of hydrogen energy and C1 chemistry in general. But it would be useful to provide capabilities along with the advantages of individual technologies. For example, lines 220 describes the benefits of a matrix convector. But it raises a number of questions:

  • What is the capacity of the matrix convector for hydrogen (synthesis gas)?
  • What are the dimensions of the convector?
  • How is the power being varied?
  • The resulting synthesis gas will have a sufficiently high nitrogen content, which may adversely affect the unit power of the devices that stand after the convector. For what processes, according to the Authors, is it most appropriate to use this technology?

At the discretion of the Authors, the same issues can be covered in more detail technical issues for other presented technologies.

Line 146 - in equation CH4

Author Response

We are gratful to the Reviewer for the careful analysis of the work and tried to take into account the comments made as much as possible. Please see the attachment. 

Reviewer 3 Report

Ref: methane-1601814-peer-review-v1
Title:  The Fuel of Our Future: Hydrogen or Methane?
Article Type:  Review

Comments to the authors

The review paper submitted to the Journal of Methane is entitled “The Fuel of Our Future: Hydrogen or Methane” It deals with the comparison of hydrogen and methane from the perspective of sustainable energy and environmental preservation. The authors reviewed well both the existing technologies and the expected one in the distant future. However, some improvements should be implemented before manuscript publication.

  • The English language should be thoroughly revised, by replacing the long statements with easily comprehendible short sentences.  For instance, lines 4-53 on page 2.
  • The hydrogen energy is described as environmentally unattractive on page 1, line 13. This assertion needs further expansion and justifications.
  • The estimates should be provided numerically, that show hydrogen energy is far lower than that of methane-based ones, the CO2 emission is higher, page 1, line 15.
  • On page 2, line 72, the authors mentioned the traditional catalytic process. It is suggested that they expand the catalytic process by describing the dry reforming and the decomposition of methane and include the research in the references. (https://doi.org/10.1016/j.2020.119445;https://doi.org/10.1016/j.cattod.2019.09.003ï¼› https://www.mdpi.com/2073-4344/10/2/242)
  • In line 73, page 2, the authors quoted the auto-thermal non-catalytic technologies. There is a need for a detailed description of the auto-thermality of the process by specifying the components used for this task (H2O, O2, CO2).
  • 6- On page 4, line 146, the authors should review and correct equation (1).
  • 7- On page 6, line 22 fat natural gases are mentioned. Authors should verify the term fat
  • 8- There could be some acronyms in the article, authors are advised to describe them before using them

Author Response

(The authors gave the same response as above.)

Round 2

Reviewer 1 Report

The authors have done the required modifications in the revised manuscript. I would like to recommend it for publication.